# Establishment and Application of Indirect ELISAs for Detecting Antibodies against Goose Astrovirus Genotype 1 and 2

**DOI:** 10.3390/vaccines11030664

**Published:** 2023-03-15

**Authors:** Mengran Zhang, Xinyu Wei, Jing Qian, Zhengyu Yu, Xin Liu, Yan Luo, Haitao Zhang, Youfang Gu, Yin Li

**Affiliations:** 1Institute of Veterinary Medicine, Jiangsu Academy of Agricultural Science, Nanjing 210014, China; 2College of Animal Science, Anhui Science and Technology University, Fengyang 233100, China; 3Key Laboratory of Veterinary Biological Engineering and Technology, Ministry of Agriculture, Nanjing 210014, China; 4GuoTai (Taizhou) Centre of Technology Innovation for Veterinary Biologicals, Taizhou 225300, China; 5Key Laboratory of Animal Diseases Diagnostic and Immunology, Ministry of Agricultural, College of Veterinary Medicine, Nanjing Agricultural University, Nanjing 210095, China; 6Jiangsu Lihua Animal Husbandry Co., Ltd., Changzhou 213168, China; 7Lihua (Nanjing) Agricultural Industry Research Institute Co., Ltd., Nanjing 210014, China

**Keywords:** goose astrovirus, GAstV-1, GAstV-2, capsid protein, antibody, indirect ELISA

## Abstract

Goose astrovirus (GAstV) was classified into GAstV-1 and GAstV-2, and both caused gosling viral gout. Recently, there has been no effective commercial vaccine to control the infection. It is important to establish serological methods to distinguish between the two genotypes. In this study, we reported the development and application of two indirect enzyme-linked immunosorbent assays (ELISAs) using the GAstV-1 virus and a recombinant GAstV-2 capsid protein as specific antigens to detect antibodies against GAstV-1 and GAstV-2, respectively. The optimal coating antigen concentration of indirect GAstV-1-ELISA and GAstV-2-Cap-ELISA was 1.2 µg/well and 125 ng/well, respectively. In addition, the antigen coating temperature and time, sera dilution and reaction time, and the dilution and reaction time of HRP-conjugated secondary antibody were optimized. The cut-off values were 0.315 and 0.305, and the analytical sensitivity was 1:6400 and 1:3200 for indirect GAstV-1-ELISA and GAstV-2-Cap-ELISA, respectively. The assays were able to differentiate specific sera against GAstVs, TUMV, GPV, and H9N2-AIV. The intra- and inter-plate variabilities of indirect ELISAs were less than 10%. The coincidence rate of positive sera was higher than 90%. The indirect ELISAs were further applied to test 595 goose serum samples. The results showed that the detection rates were 33.3% and 71.4% in GAstV-1-ELISA and GAstV-2-Cap-ELISA, respectively, and the co-detection rate was 31.1%, which indicates that the seroprevalence rate of GAstv-2 was higher than that of GastV-1, and the co-infection existed between GAstV-1 and GAstV-2. In summary, the developed GAstV-1-ELISA and GAstV-2-Cap-ELISA have high specificity, sensitivity, and reproducibility and can be used in the clinical detection of the antibody against GAstV-1 and GAstV-2.

## 1. Introduction

Goose astrovirus (GAstV) caused an infectious disease characterized by visceral and joint urate deposition and the hemorrhage and swelling of kidneys, mainly infected 1- to 20-day-old goslings, resulting in the rate of infection and mortality up to 80% and 50% in some flocks, respectively, which caused huge economic losses to the goose industry [1,2,3]. GAstVs are non-enveloped, single-stranded, positive-sense RNA viruses, a member of the family *Astroviridae*, genus *Avastroviruses* [4]. The complete GAstV genome length is 6.8–7.9 kb, consisting of a 5′-untranslated region (UTR), three opening reading frames (ORF1a, ORF1b, and ORF2), a 3′-UTR, and a poly-A tail [5]. ORF1a and ORF1b encode non-structural proteins, and ORF2 encodes a capsid protein, which contains a conserved N-terminal capsid core and a highly variable C-terminal spike domain [6]. The spike domain contains receptor binding regions and the virus-neutralizing antibodies binding sites, which determine the host range and specificity of the virus [7]. Thus, capsid protein plays an important role as an immunogen that stimulates the host to produce neutralizing antibodies [7]. In addition, GAstV-1 and GAstV-2 share 40.4%–41.6% ORF2 nucleotide sequence identity [5]. Thus, GAstV can be divided into two genotype groups, GAstV-1 and GAstV-2, based on the nucleotide sequences of the ORF2 region [8].

Several GAstV strains have been reported and sequenced in the east of China, including the provinces of Shandong, Jiangsu, Anhui, Zhejiang, Fujian, and Shanghai [8,9,10]. In addition, the data showed that infections of GAstV-1 and GAstV-2 were detected in Jiangsu province [11,12], which presents a challenge to the timely diagnosis and control of co-infection with GAstV-1 and GAstV-2. Currently, there is no effective commercial vaccine to prevent and control GAstV infection. There were several diagnostic techniques available to detect early GAstV infection, including reverse transcription polymerase chain reaction (RT-PCR) [13], loop-mediated isothermal amplification [14], and quantitative fluorescence PCR [15]. Most of them have focused on the detection of GAstV-2 infection. In 2022, a duplex TaqMan real-time RT-PCR assay for simultaneous detection of GAstV-1 and 2 were established [16]. In addition, available serological methods for the detection of antibodies against GAstV, including indirect immunofluorescence (IFA), enzyme-linked immunosorbent assay (ELISA), viral neutralization assay (VN), and Agar Gel Immunodiffusion (AGID) [17,18,19,20,21]. However, there is no serological method for distinguishing between the antibodies against GastV-1 and GAstV-2. Accordingly, a novel method for the simultaneous detection of the antibodies against GAstV-1 and GAstV-2 provides useful serological assays for epidemiological investigation.

## 2. Materials and Methods

### 2.1. Virus, Cells and Serum Samples

The GAstV-2 AHQJ18 (Genbank Accession: OP556137) and GAstV-1 JS33-3 (Genbank Accession: OP272633) were isolated from goslings with clinical gout in Anhui and Jiangsu Provinces, respectively. *Spodoptera frugiperda* (Sf9) cells (Invitrogen Corporation, CA, USA) were used to rescue a recombinant baculovirus in Grace’s Insect Cell Culture Medium (Thermo Fisher Scientific, Shanghai, China), supplemented with 10% fetal bovine serum (Wisent, Nanjing, China) and 100 U/mL streptomycin-penicillin (Thermo Fisher Scientific). High Five (Hi5) cells (Sunncell) were used for large-scale expression of the recombinant protein, which propagated in the IB905 Insect culture medium (Innovative Bioscience Co., Ltd., Beijing, China). The mouse anti-ORF2 polyclonal antibody and the sera against GAstV-1, GAstV-2, goose parvovirus (GPV), goose Tembusu virus (TUMV), and H9N2 avian influenza virus (H9N2-AIV) were preserved in our lab, Laboratory of Avian Disease, Institute of Veterinary Medicine, Jiangsu Academy of Agricultural Sciences in China. The mouse anti-flag monoclonal antibody was purchased from Beyotime (Shanghai, China). Positive and negative serum samples used in our study were collected by Jiangsu Lihua Animal Husbandry Co., Ltd.

### 2.2. Expression and Purification of GAstV-2 ORF2 Protein

To express the ORF2 gene in insect cells, which was amplified from the GAstV-2 AHQJ18 and inserted into the baculovirus expression vector pFastBac1 (Beyotime). The ORF2 gene sized 2115 bp was amplified by PCR from cDNA of LMH cells culture supernatants infected with the GAstV-2 AHQJ18, using ORF2-F1 and ORF2-R1 (Table 1). Sequences were confirmed by sequence analysis. Subsequently, ORF-F2 and ORF2-R2 (Table 1) were used to amplify the homologous sequences with the PCR products as templates to insert the target gene and Flag-tag into the multiple cloning sites between the *Not* I and *Xho* I restriction sites downstream from the polyhedron promoter (P_PH_) of the pFastBac1 vector (Figure 1). Sequences were confirmed by sequence analysis. The recombinant plasmid was transferred into competent *Escherichia coli* DH10Bac cells to generate recombinant bacmid DNA (rBacmid-ORF2), according to the manufacturer’s instruction of the Bac-to-Bac Baculovirus Expression System (Beyotime). The rBacmid-ORF2 was verified by PCR analysis using M13 universal primers (Table 1) to confirm the insertion of the ORF2 gene. The rBacmid-ORF2 was transfected into Sf9 cells using Lipofectamine^®^ 3000 reagent (Thermo Fisher Scientific) to construct the recombinant virus for protein expression. After 72 h post-transfection in the 27 °C incubator, the P0 virus was recovered, and the recombinant baculovirus virus carrying the ORF2 gene (rBV-ORF2) was obtained. The specificity of the rBV-ORF2 was determined by Western blotting using the mouse anti-ORF2 polyclonal antibody and anti-flag monoclonal antibody. Subsequently, rBV-ORF2 was infected with Hi5 cells in incubation at 27 °C for 72 h. The suspension was collected after centrifugation at 5000× *g* for 20 min and purified using anti-flag affinity gel (Beyotime) according to the manufacturer’s instructions to harvest the recombinant protein (rCap). Finally, the purity was assessed by 10% sodium dodecyl sulfate-polyacrylamide gel electrophoresis (SDS-PAGE, Genscript, Nanjing, China) and analyzed using ImageJ software.

### 2.3. Preparation and Purification of GAstV-1

The GAstV-1 JS33-3 was isolated and proliferated using 9-day-old goose embryos in a humidified incubator at 37 °C. Allantoic fluid was collected from 24 h to 144 h dead embryos and purified using the differential centrifugation technique [15]. Briefly, the allantoic fluid was centrifuged at 1200× *g* for 1 h to retain the supernatant, then centrifuged at 40,000× *g* for 2 h to harvest the precipitate, which was diluted and dissolved by sterile phosphate-buffered saline (PBS, pH 7.4) and stored at −20 °C. The GAstV-1 was inactivated by formaldehyde (Sinopharm chemical reagent Co., Ltd., Beijing, China) at a final concentration of 1/1000 and used as the coating virus for GAstV-1-ELISA.

### 2.4. SDS-PAGE and Western Blotting Analysis

The rBV-ORF2 was verified by Western blotting assay. The clarified lysate from Sf9 cells inoculated with rBV-Cap was separated by 10% SDS-PAGE and transferred to a polyvinylidene fluoride (PVDF) membrane (Bio-Rad Laboratories Srl, Milan, Italy). The membrane was blocked with 5% (*w*/*v*) nonfat milk in PBST (PBS containing 0.05% Tween-20) for 2 h at 37 °C and incubated with the mouse anti-Cap polyclonal antibody or mouse anti-flag monoclonal antibody overnight at 4 °C to probe target protein. After 3 washes in PBST for 5 min each, the membranes were incubated with secondary horseradish peroxidase (HRP)-conjugated goat anti-mouse antibodies (Beyotime). The reaction was visualized by DAB Horseradish Peroxidase Color Development Kit (Beyotime) or a chemiluminescence assay (ECL, Beyotime).

### 2.5. Development of GAstV-1-ELISA and GAstV-2-Cap-ELISA

Optimum dilutions of antigens and sera were determined by a checkerboard titration test using known goose positive and negative sera. The protein and virus were diluted by double dilution method with carbonate-bicarbonate buffer (pH 9.6) and coated in 96-well plates (Corning, NY, USA) at 4 °C overnight. After washing 3 times with PBST and patting dry, the plates were blocked for 2 h at 37 °C with 5% nonfat milk and washed as above. Then, each well was added 100 µL diluted sera by double dilution method and incubated at 37 °C for 1 h. After extensive washing, the 1:1000 HRP-conjugated goat anti-goose IgG (preserved in our lab) was added and incubated at 37 °C for 1 h. 100 µL 3,3′,5,5′-tetramethylbenzidine (TMB, Beyotime) was added in each well and incubated at room temperature for 10 min after the plates were washed. The reaction was stopped using 50 µL 2 mol/L H_2_SO_4_. The optical density (OD) of each well was measured at 450 nm using a spectrophotometer (BioTek, Winooski, VT, USA). The dilutions with the maximum difference in absorbance at 450 nm (OD_450_ nm) between the positive and the negative sera (P/N) were selected to optimize the following reaction conditions, including coating conditions, serum reaction time, and HRP-conjugated secondary antibody dilution and reaction time.

### 2.6. Determination of the Cut-off Value for GAstV-1-ELISA and GAstV-2-Cap-ELISA

Forty negative sera were used to determine the cut-off values of the indirect ELISAs, conducted in optimal conditions. The mean (X) and standard deviation (SD) of the sera OD_450_ nm were analyzed. The cut-off values were determined as X + 3SD. The OD_450_ nm of serum samples ≥ X + 3SD were positive, otherwise negative.

### 2.7. Determination of the Specificity, Sensitivity, and Repeatability of GAstV-1-ELISA and GAstV-2-Cap-ELISA

The specificity of the established GAstV-1-ELISA and GAstV-2-Cap-ELISA were determined by testing positive sera against GAstV-1, GAstV-2, TUMV, GPV, and H9N2-AIV and negative serum.

Three GAstV-1-positive and 3 GAstV-2-positive sera were used to assess the sensitivity of the GAstV-1-ELISA and GAstV-2-Cap-ELISA, respectively. The positive sera were diluted to 1:100, 1:200, 1:400, 1:800, 1:1600, 1:3200, 1:6400 and 1:12,800.

Six sera were tested on a plate by an analyst, 3 replicates, to determine the repeatability (intra-plate variability). The mean, standard deviation, and coefficients of variation (CVs, CV = SD/mean × 100%) were computed using Excel 2019 (Microsoft Corporation, Redmond, WA, USA).

Three 96-well plates, coated with distinct batches of antigens, were used to test the same 6 sera in triplicates and calculated the mean and standard deviation of each sample’s OD_450_ nm. The replicability (inter-plate variability) between plates was assessed by CVs.

### 2.8. Comparison of Indirect ELISAs with AGID Assay

Forty GAstV-1-positive and forty GAstV-2-positive sera (preserved in our lab) were used to compare the detection rate and coincidence rate between the established indirect ELISAs and AGID assay. Our previous studies showed that the AGID assay could determine the specific binding between antigens and antibodies [19]. In brief, the antigen was concentrated 50 times by ultracentrifugation and deposited in the center well, and the tested serum sample was added to the surrounding wells. After 24–48 h incubation, if the center well and surrounding wells interacted, a precipitation line will be observed, which indicates the tested serum samples are positive.

### 2.9. Application of GAstV-1-ELISA and GAstV-2-Cap-ELISA in Clinical Samples

A total of 595 clinical goose serum samples from Suqian (*n* = 345), Changzhou (*n* = 156), and Zhenjiang (*n* = 94) goose farms in Jiangsu Province in 2022 were tested by GAstV-1-ELISA and GAstV-2-Cap-ELISA for the detection of antibodies against GAstV and co-seroprevalence of GAstV-1 and GAstV-2.

### 2.10. Statistical Analysis

GraphPad Prism 9.4.1 software was used for statistical analysis and figure drawing. All data are presented as the mean or the mean ± standard deviation (SD). Student’s *t*-test was used to assess differences in the experimental data. Statistical significance set at *p* < 0.05.

## 3. Results

### 3.1. Expression and Purification of the Recombinant Protein

Following digestion, ligation and sequencing, the rBacmid-ORF2 was transfected into Sf9 cells to rescue rBV-ORF2. Western blotting results showed that a target band of 86 kDa was found, but no band could be detected for the empty vector, indicating that the rCap was successfully expressed and the expressed band had a superior immunoreaction with mouse anti-ORF2 polyclonal antibodies and mouse anti-flag monoclonal antibodies (Figure 2A,B). Anti-flag affinity gel was applied to purify the Flag-tagged rBV-ORF2, and a single band was detected by SDS-PAGE (Figure 2C), and the purity is 97%. The concentration of rCap estimated using a BCA protein quantification kit (Vazyme, Nanjing, China) was 0.27 mg/mL.

### 3.2. Establishment of Indirect ELISAs

#### 3.2.1. Establishment of Indirect ELISA for GAstV-1 (GAstV-1-ELISA)

The checkerboard titration results showed the optimal coating virus concentration was 1.2 µg/well, and sera dilution was 1:400 (Appendix A). The coating virus was coated at 4 °C overnight (Figure 3A). The diluted sera reacted at 37 °C for 1.5 h (Figure 3B). The HRP-conjugated secondary antibody was diluted at 1:800 (Figure 3C) and reacted at 37 °C for 1 h (Figure 3D). The mean value and SD were 0.188 and 0.0423, respectively, and the cut-off value of GAstV-1-ELISA was 0.315 (Figure 3E).

#### 3.2.2. Establishment of Indirect ELISA for GAstV-2 (GAstV-2-Cap-ELISA)

The checkerboard titration results showed the optimal coating protein concentration was 125 ng/well, and the sera dilution was 1:200 (Appendix A). The protein was coated at 4 °C overnight (Figure 4A), the diluted sera reacted at 37 °C for 1.5 h (Figure 4B), and the HRP-conjugated secondary antibody was diluted at 1:500 (Figure 4C) and reacted at 37 °C for 1 h (Figure 4D). The mean value and SD were 0.138 and 00423, respectively, and the cut-off value of GAstV-2-Cap-ELISA was 0.305 (Figure 4E).

### 3.3. Determination of the Specificity, Sensitivity and Repeatability of GAstV-1-ELISA and GAstV-2-Cap-ELISA

The positive sera against GAstV-1, GAstV-2, TUMV, GPV, and H9N2-AIV and negative sera were used to detect the specificity of GAstV-1-ELISA and GAstV-2-Cap-ELISA. The results showed that the tested sera were GAstV-seronegative with the exception of the GAstV-1 and GAstV-2 positive sera for GAstV-1-ELISA and GAstV-2-Cap-ELISA, respectively (Figure 5A,C).

The sensitivity of GAstV-1-ELISA and GAstV-2-Cap-ELISA were determined using three corresponding positive sera, serially diluted two-fold from 1:100 to 1:12,800. The results showed that the minimum detection limit of GAstV-1-ELISA and GAstV-2-Cap-ELISA were 1:6400 and 1:3200, respectively (Figure 5B,D).

Six sera were randomly selected to assess the repeatability of GAstV-1-ELISA and GAstV-2-Cap-ELISA. The CVs ranged from 1.3% to 8.7% between intra- and inter-plate variability for GAstV-1-ELISA (Table 2). For GAstV-2-Cap-ELISA, the CVs ranged from 1.1% to 7.3% (Table 2).

### 3.4. Comparison of Indirect ELISAs and the AGID Assay

Forty GAstV-1-positive sera were used to assess the potential of developed GAstV-1-ELISA by comparison with AGID assay. The results showed that the detection rate of GAstV-1-ELISA was 92.5% (37/40) and that of AGID assay was 52.5% (21/40), indicating that GAstV-1-ELISA was more sensitive than AGID, and the developed GAstV-1-ELISA showed 60% (24/40) coincidence rate with the AGID assay (Table 3).

Forty GAstV-2-positive sera were detected by GAstV-2-Cap-ELISA and AGID assay. The results indicated that the detection rate of GAstV-2-Cap-ELISA and AGID assay was 100% (40/40) and 65% (26/40), respectively. The coincidence rate of the assays was 65% (26/40), indicating that GAstV-2-Cap-ELISA is more sensitive than AGID assay (Table 3).

### 3.5. Application of GAstv-1-ELISA and GAstV-2-Cap-ELISA in Detecting Antibodies

The results showed that the seroprevalence rate of GAstV-1 from Suqian, Changzhou, and Zhenjiang serum samples were 2.3%, 83.3%, and 63.8%, respectively, and that of GAstV-2 were 58.8%, 93.6%, and 80.9%. In addition, the co-seroprevalence of GAstV-1 and GAstV-2 was 2.3%, 78.8%, and 57.4% in Suqian, Changzhou, and Zhenjiang serum samples, respectively. In general, the co-seroprevalence of GAstV-1 and GAstV-2 was 31.1%, and the seroprevalence rate of GAstV-1 and GAstV-2 was 33.3% and 71.4% in 595 clinical goose serum samples from Jiangsu province, respectively (Table 4).

## 4. Discussion

GAstV was first isolated in 2016 and had spread rapidly in serval provinces in China [1,22]. Studies of the GAstV pathogenicity showed that the incidence of goslings reached 100% at different days of age, and the infection caused growth repression, severe visceral urate deposition, and even death in 5- and 15-day-old goslings, whereas goslings infected at 25- and 35-day-old showed mild symptoms, which had also been confirmed in histopathological experiments [23]. To control GAstV infection, numerous diagnostic methods have been developed, including virus isolation and identification and serological and molecular biological detection. GAstV was successfully isolated from goose embryos, LMH cells, chicken embryo fibroblast cells, and goose embryo kidney cells [2,3]. However, our previous studies found a difference in biological characteristics between GAstV-1 and GAstV-2. GAstV-1 could be multiplied using goose embryos, while it was difficult to amplify a high content of GAstV-2 and only lasted for five generations (data unpublished). In addition, some studies found GAstV-2 could be propagated in LMH cells, but there was no apparent cytopathic effect (CPE) [24]. In addition, pathogen isolation of infected geese revealed that GAstV existed the co-infection with other viruses, such as novel duck reovirus, GPV, and TMUV [16,25,26,27], and caused immune suppression in infected geese, resulting in susceptibility to bacterial and fungal infections [28]. Wang developed a duplex real-time RT-PCR assay for the simultaneous detection of GAstV-1 and GAstV-2 [12]. Wang established an indirect ELISA based on a core antigenic advantage domain of capsid protein (shCap) to detect GAstV-2 antibodies, indicating that shCap could be used as a coating protein to capture antibodies [17]. Currently, there is no serological diagnostic method to distinguish between GAstV-1 and GAstV-2 infection.

Indirect ELISA is simple, sensitive, and field-deployed, which is significant for rapid diagnosis and elimination of infectious sources and interruption of the infection of GAstV [29]. In this study, forty known positive serum samples were tested by AGID assay and the developed indirect ELISAs. The results showed that the detection rate of indirect ELISAs was more sensitive and the detection rate reached more than 90%, and the coincidence rate of the two methods was about 60%. Moreover, we tested 595 field serum samples from serval goose farms in Jiangsu province. The results showed that the co-seroprevalence of GAstV-1 and GAstV-2 was 31.1%, and the seroprevalence rate of GAstV-1 and GAstV-2 was 33.3% and 71.4% in 595 clinical goose serum samples from Jiangsu province, respectively, which indicating the co-infection of two genotypes has existed, and the GAstV-2 is an endemic strain in Jiangsu. However, in this study, we only used serum samples from Jiangsu province, which are not large enough to analyze the accurate epidemiological status of GAstV in China. An extensive serological survey is urgent to evaluate the prevalence and develop reasonable prevention and control measures.

The developed indirect ELISAs can be applied to the detection of antibodies against GAstV in all ages of geese. At the same time, the most important tested subjects are 1–35-day-old goslings and breeding geese owing to the transmission route of the infection and pathogenicity of GAstV. Vertical transmission is a critical transmission route of GAstV infection; latently infected breeding geese pass the virus to the next generation through eggs [2]. Breeding geese with were isolated and raised into two groups, the negative group and the antibody group, by ELISA detecting the level of antibodies in vivo. One-day-old goslings hatched from the antibody group achieved the maternal antibody and then decreased a week later. In addition, 35-day-old goslings were tested by ELISAs, and if the antibody level increased again, the goslings were infected. The established ELISA is conducive to the purification of GAstV in goose farms.

## 5. Conclusions

In this study, we established two indirect ELISAs to detect the antibody against GAstVs-1 and GAstV-2. The indirect GAstV-1-ELISA was established using the GAstV-1 virus, and GAstV-2-Cap-ELISA was developed using a recombinant GAstV-2 capsid protein expressed using the baculovirus expression system. The results showed that the established indirect ELISAs are specific and sensitive, and there was no cross-reaction between the GAstV-1 and GAstV-2 genotypes.

## Figures and Tables

**Figure 1 vaccines-11-00664-f001:**
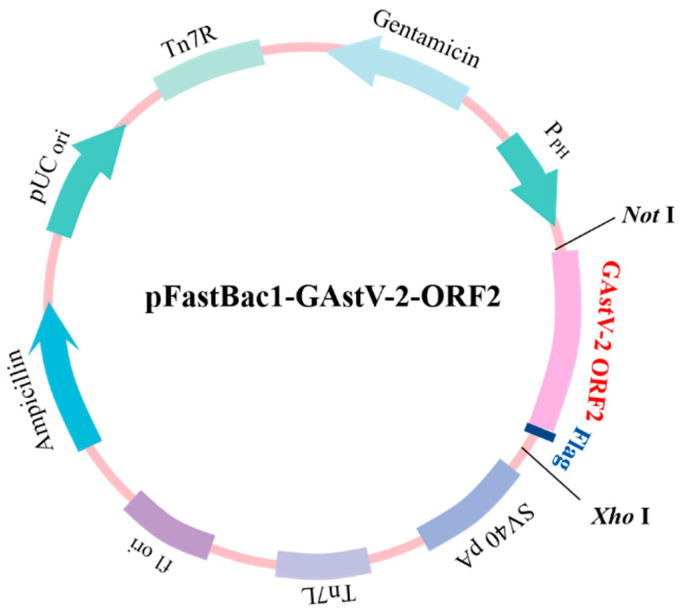
Schematic diagram of pFastBac1 vector construction. The GAstV-2 ORF2 gen and Flag-tag were inserted into the multiple cloning sites between the *Not* I and *Xho* I restriction sites downstream of the P_PH_ of the pFastBac1 vector.

**Figure 2 vaccines-11-00664-f002:**
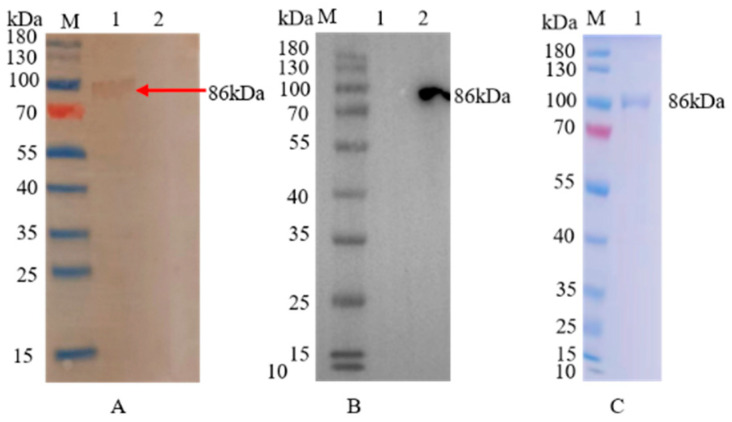
Expression and purification of rCap. (**A**) Western blotting analysis of rBV-ORF2 with mouse anti-ORF2 polyclonal antibodies. M: Protein molecular marker; Lane 1: the Sf9 cells transfected with rBacmid-ORF2; Lane 2: the Sf9 cells transfected with rBacmid control. (**B**) Western blotting analysis of rBV-ORF2 with mouse anti-flag monoclonal antibodies. M: Protein molecular marker; Lane 1: the Sf9 cells transfected with rBacmid control; Lane 2: the Sf9 cells transfected with rBacmid-ORF2; (**C**) The SDS-PAGE analysis of purified rCap. M: Protein molecular marker; Lane 1: purified rCap.

**Figure 3 vaccines-11-00664-f003:**
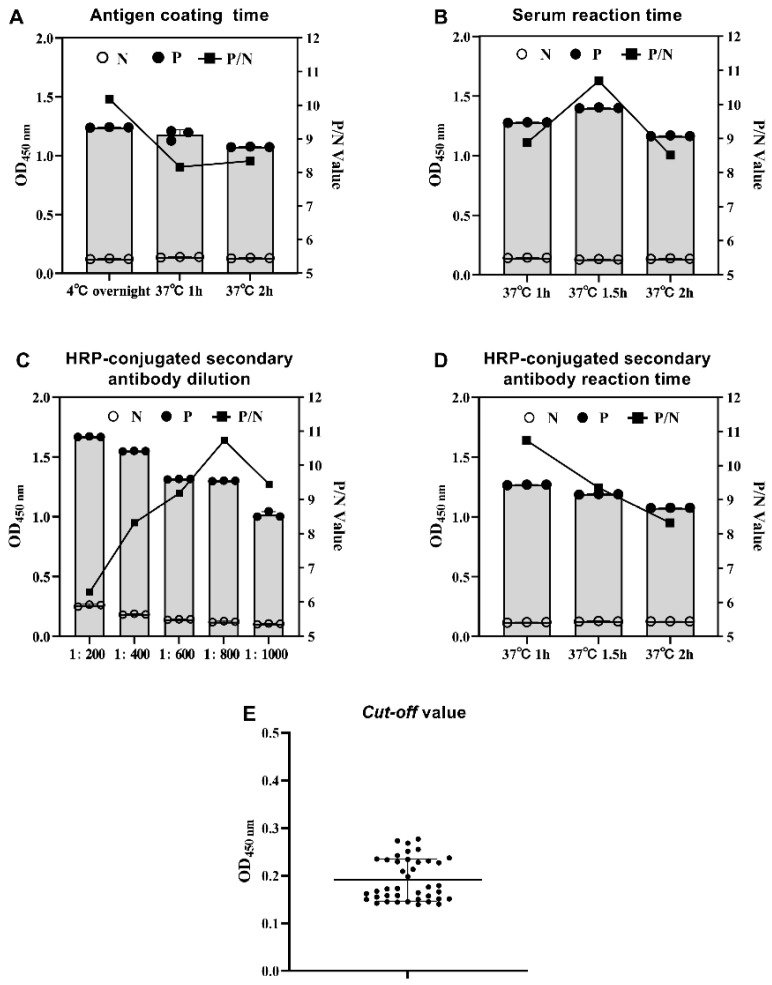
Optimization of GAstV-1-ELISA reaction conditions. (**A**) Optimization of virus coating conditions. ELISA plates with 1.2 µg/well coating virus were incubated at 4 °C overnight, 37 °C for 1 h, and 37 °C for 2 h. (**B**) Optimization of reaction time of sera. The sera with dilutions of 1:400 was reacted at 37 °C for 1 h, 1.5 h, and 2 h. (**C**) Optimization of HRP-conjugated secondary antibody dilution. The dilution of the HRP-conjugated secondary antibody was 1:200, 1:400, 1:600, 1:800, and 1:1000. (**D**) Optimization of HRP-conjugated secondary antibody reaction time. The HRP-conjugated secondary antibody was reacted at 37 °C for 1 h, 1.5 h, and 2 h. (**E**) The determination of GAstV-1-ELISA cut-off value. Forty negative sera were used to calculate the cut-off value of GAstV-1-ELISA.

**Figure 4 vaccines-11-00664-f004:**
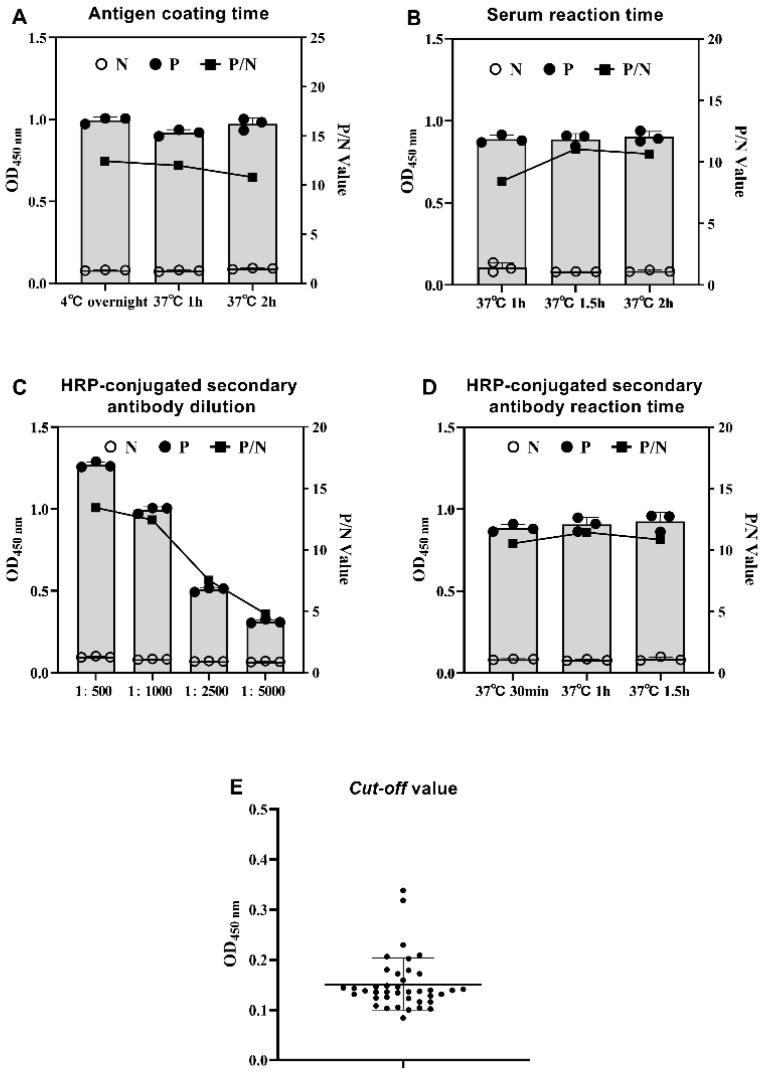
Optimization of GAstV-2-Cap-ELISA reaction conditions. (**A**) Optimization of protein coating conditions. ELISA plates with 125 ng/well coating protein were incubated at 4 °C overnight, 37 °C for 1 h, and 37 °C for 2 h. (**B**) Optimization of reaction time of sera. The sera with dilutions of 1:200 were reacted at 37 °C for 1 h, 1.5 h, and 2 h. (**C**) Optimization of dilution of HRP-conjugated secondary antibody. The dilution of the HRP-conjugated secondary antibody was 1:500, 1:1000, 1:2500, and 1:5000. (**D**) Optimization of reaction time of HRP-conjugated secondary antibody. The HRP-conjugated secondary antibody was reacted at 37 °C for 30 min, 1 h, and 1.5 h. (**E**) The determination of GAstV-2-Cap-ELISA cut-off value. Forty negative sera were used to calculate the cut-off value of GAstV-2-Cap-ELISA.

**Figure 5 vaccines-11-00664-f005:**
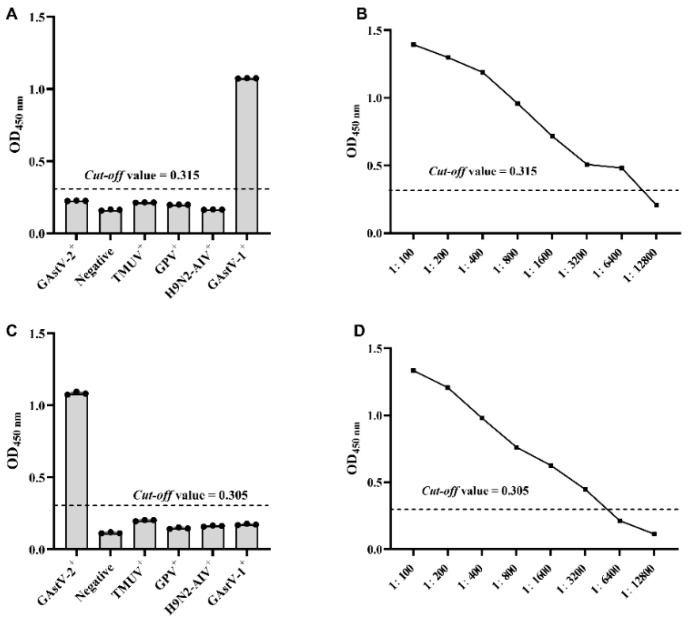
Specificity and sensitivity of GAstV-1-ELISA and GAstV-2-Cap-ELISA. (**A**) The specificity of GAstV-1-ELISA. The OD_450_ nm was determined using positive serum samples against GAstV-1, GAstV-2, TMUV, GPV, and H9N2-AIV and negative serum. (**B**) The sensitivity of GAstV-1-ELISA. The positive sera were diluted to 1:100, 1:200, 1:400, 1:800, 1:1600, 1:3200, 1:6400, and 1:12,800. (**C**) The specificity of GAstV-2-Cap-ELISA. The OD_450_ nm was determined using positive serum samples against GAstV-1, GAstV-2, TMUV, GPV, and H9N2-AIV and negative serum. (**D**) The sensitivity of GAstV-2-Cap-ELISA. The positive sera were diluted to 1:100, 1:200, 1:400, 1:800, 1:1600, 1:3200, 1:6400, and 1:12,800.

**Table 1 vaccines-11-00664-t001:** Sequences of primers used in this study.

Primer Name	Primer Sequence (5′-3′)
ORF2-F1	ATGGCAGACAGGGCGGTGG
ORF2-R1	TCACTTGTCATCGTCGTCCTTGTAATCCTCATGTCCGCCCTTC ^a^
ORF2-F2	GACGACCTCACTAGTCGCGGCCGCATGGCAGACAGGGCGG ^b^
ORF2-F2	GCTTGGTACCGCATGCCTCGAGTCACTTGTCATCGTCGTC ^c^
M13-F	CCCAGTCACGACGTTGTAAACG
M13-R	AGCGGATAACAATTTCACACAGG

^a^ In italics is the flag-tag sequence. ^b^ Underlined is Not I restriction site. ^c^ Underlined is the Xho I restriction site.

**Table 2 vaccines-11-00664-t002:** Intra- and inter-plate repeatability test of GAstv-1-ELISA and GAstV-2-Cap-ELISA.

Serum Samples	GAstV-1-ELISA				GAstV-2-Cap-ELISA			
	Intra-Assay		Inter-Assay		Intra-Assay		Inter-Assay	
	X ± SD	CV%	X ± SD	CV%	X ± SD	CV%	X ± SD	CV%
1	0.253 ± 0.006	2.4	0.249 ± 0.019	7.8	0.463 ± 0.005	1.1	0.447 ± 0.023	5.2
2	0.493 ± 0.007	1.3	0.512 ± 0.020	3.9	0.257 ± 0.010	4.0	0.217 ± 0.014	6.6
3	0.186 ± 0.011	5.9	0.162 ± 0.014	8.7	0.117 ± 0.004	3.2	0.146 ± 0.011	7.3
4	0.624 ± 0.008	1.3	0.626 ± 0.017	2.7	0.636 ± 0.008	1.3	0.666 ± 0.036	5.3
5	0.942 ± 0.027	2.8	0.952 ± 0.032	3.4	1.118 ± 0.024	2.2	1.114 ± 0.024	2.2
6	0.849 ± 0.027	3.2	0.840 ± 0.020	2.4	0.439 ± 0.027	6.1	0.447 ± 0.021	4.8

**Table 3 vaccines-11-00664-t003:** Comparison of indirect ELISAs and AGID assay.

AGID	GAstV-1-ELISA				GAstV-2-Cap-ELISA			
	Positive (*n*)	Negative (*n*)	Total (*n*)	Coincidence Rate (%)	Positive (*n*)	Negative (*n*)	Total (*n*)	Coincidence Rate (%)
Positive	21	0	21	60 (24/40)	26	0	26	65 (26/40)
negative	16	3	19		14	0	14	
Total	37	3	40		40	0	40	

**Table 4 vaccines-11-00664-t004:** GAstVs antibody detection results by the established ELISAs from 595 serum samples.

	GAstV-1-ELISA			GAstV-2-Cap-ELISA		Co-Positivity (%)
	Samples (*n*)	Positive (*n*)	Positive Rate (%)	Positive (*n*)	Positive Rate (%)	
Suqian	345	8	2.3 (8/345)	203	58.8 (203/345)	2.3 (8/345)
Changzhou	156	130	83.3 (130/156)	146	93.6 (146/156)	78.8 (123/156)
Zhenjiang	94	60	63.8 (60/94)	76	80.9 (76/94)	57.4 (54/94)
Total	595	198	33.3 (198/595)	425	71.4 (425/595)	31.1 (185/595)

## Data Availability

Not applicable.

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
