# Peer review of "Establishment and Application of Indirect ELISAs for Detecting Antibodies against Goose Astrovirus Genotype 1 and 2"

_vaccines, 2023, doi:10.3390/vaccines11030664_

Round 1

Reviewer 1 Report

The authors present a strategy for the implementation of two in house ELISA tests. The methodology and implementation for this method is well described and the results are presented properly. 

Considering what is mentioned in lines 60-65. I suggest to review this paragraph and separate the use of methods for the antigen or antibodies detection. These will help to close the paragraph and to clarify the final statement: "By testing the serum samples for detecting potential virus-carrying 66 animals in epidemic situations." Lines 66-67. 

It would be useful to comment about samples which results are in the grey zone according to OD reading and the cut off values that were established.

In terms of background and relevant references, there are recent publications regarding molecular characterization of GAstVs that would be useful to include. Therefore it is suggested to update and to include recent references.

Please review and complete Reference information that is missing. The year of publication is missing in several references ( 4, 5, 6, 7, 9, 10, 11...)

The authors present the approach and results to determine specificity, sensibility and  repeatability of the test. However, there is not information about robustness and uncertainty, which will be useful if it is intended to use these tests for serological diagnosis.

Please indicate how many replicates of the three positive and three negative serum samples were used to determine the sensitivity of the test.

Considering the sensibility and other characteristics own to the AGID test it is not the best to compare  with the ELISAs proposed in the manuscript. It would be useful to use virus neutralization instead. Please justify the use of AGID and how it supports properly the results. 

Author Response

Thanks for your letter and for reviewers’ comments concerning our manuscript entitled “Establishment and Application of two indirect ELISAs for detecting antibodies against GAstV-1 and GAstV-2” (Manuscript ID: vaccines- 2263633). Those comments are all valuable and helpful for revising and improving our paper. We have studied all comments carefully and have made conscientious corrections. Revised portions are marked in the paper. The main corrections in the paper and the responses to the reviewers’ comments are as follows.

Reviewer 2 Report

Please find the attached file for comments to authors

Reviewer 3 Report

Zhang and colleagues describe the establishment of ELISA assays for the detection of antibodies against goose astrovirus 1 and 2 in goose. For GAstrV-1 they use full virus antigens obtained by cell culture while for GAstrV-2 they use a recombinant cap protein fragment produced in a baculovirus insect cell system. They describe in very detail the methods used and reveal the both ELISAs show very good quality properties and, most important, no cross-reactivities between both astrovirus genotypes.

The paper is well written. There are only some minor remarks:

Title: please spell in full the virus names in the title of the paper

Line 45: Could you please add some information on the occurrence of GAstrV outside China?

Line 53: Please rephrase the sentence “...interact with the host cell receptor, antigenic variation, and ...”

Line 57: Genotype group: could you please add some more information: how large are the genetic differences? For example, the identities of the cap protein genes?

Lines 59-66: can these tests differ between GAstV-1 and GAstV-2? Could you please add a note on this?

Line 147: how long was the incubation time of the sera?

Line 161: how was the negative status of the 40 sera defined?

Line 166 etc: How was the status of the sera as negative or positive for GAstr-1, GAstV-2, TUMV, GPV, H9N2-AIV defined? Please spell the last three virus names in full.

Line 167: please replace “detecting” with “testing”

Line 181: How were the 40 GAstV-1-positive and 40 GAstV-2 positive sera prepared? Please add some more information.

Lines 181-188: please add some information on the antigens used in AGID

Line 306: please add some information on the occurrence outside China.

Lines 310-311: please add the diagnostic tests used in the epidemiological investigations.

 Line 318: Could you please add some information which serological methods are used for routine diagnosis of GAstV?

Lines 328-330: Did Wang et al use a GAstV-1 strain or a GAstV-2 strain in their ELISA?

Please check the reference list: the publication year is not given in several references.
